# UPF1-Mediated RNA Decay—Danse Macabre in a Cloud

**DOI:** 10.3390/biom10070999

**Published:** 2020-07-04

**Authors:** Daria Lavysh, Gabriele Neu-Yilik

**Affiliations:** 1Department of Pediatric Oncology, Hematology and Immunology, University of Heidelberg, Im Neuenheimer Feld 430, 69120 Heidelberg, Germany; daria.lavysh@med.uni-heidelberg.de; 2Molecular Medicine Partnership Unit, University of Heidelberg and European Molecular Biology Laboratory, Im Neuenheimer Feld 350, 69120 Heidelberg, Germany; 3Department Clinical Pediatric Oncology, Hopp Kindertumorzentrum am NCT Heidelberg, 69120 Heidelberg, Germany

**Keywords:** UPF1, nonsense-mediated RNA decay, UPF1-mediated RNA decay, RNA-binding protein, RNA turnover and surveillance

## Abstract

Nonsense-mediated RNA decay (NMD) is the prototype example of a whole family of RNA decay pathways that unfold around a common central effector protein called UPF1. While NMD in yeast appears to be a linear pathway, NMD in higher eukaryotes is a multifaceted phenomenon with high variability with respect to substrate RNAs, degradation efficiency, effector proteins and decay-triggering RNA features. Despite increasing knowledge of the mechanistic details, it seems ever more difficult to define NMD and to clearly distinguish it from a growing list of other UPF1-mediated RNA decay pathways (UMDs). With a focus on mammalian NMD, we here critically examine the prevailing
NMD models and the gaps and inconsistencies in these models. By exploring the minimal requirements for NMD and other UMDs, we try to elucidate whether they are separate and definable pathways, or rather variations of the same phenomenon. Finally, we suggest that the operating principle of the UPF1-mediated decay family could be considered similar to that of a computing cloud providing a flexible infrastructure with rapid elasticity and dynamic access according to specific user needs.

## 1. Nonsense-Mediated RNA Decay

Nonsense-mediated RNA decay (NMD) was originally discovered as a medically relevant cellular pathway that degrades mRNAs containing nonsense mutations in their open reading frames (ORFs), thus preventing the expression of truncated proteins. This discovery inspired the hypothesis that NMD has evolved as a cellular surveillance system to protect organisms from deleterious effects of such potentially harmful polypeptides. However, NMD is Janus-faced in the sense that it can also result in loss-of-function phenotypes when mRNAs encoding (partially) functional truncated proteins are degraded [1,2,3,4]. Furthermore, researchers increasingly have realized that both the spectrum and the sheer number of NMD-targeted mRNAs extends far beyond the relatively few transcripts that contain genomic nonsense mutations. Since several excellent reviews on various aspects of NMD have been published in the past few years [3,5,6,7,8,9,10,11,12,13,14,15,16], we here review only the information necessary to understand the rationale of this article.

NMD targets can be roughly subdivided into two groups, namely (1) erroneous transcripts and (2) error-free transcripts. Nonsense mutations introduce a translation termination codon (TC) into the ORF of an mRNA and can arise from spontaneous mutations, faulty transcription, splice errors, unproductive genomic rearrangement and unprogrammed ribosomal frame shifting, thus generating mRNAs that are detected and degraded by NMD in its function as a “vacuum cleaner” [17]. Such flawed TCs have been dubbed premature termination codons (PTCs). However, in a more sophisticated role, NMD also acts to fine-tune the expression of a plethora of error-free physiological transcripts. Proteins encoded by such transcripts are often positioned at central nodes of biochemical pathways, thus tuning the pathways’ function, shaping the expression of whole cohorts of downstream effectors, and regulating important physiological processes such as cellular stress, development and embryogenesis [5,13,18,19,20,21,22]. Such error-free NMD substrates include transcripts with upstream open reading frames (uORFs), with splice events in their 3′ untranslated regions (3′UTR) generated by regular alternative splicing, with exceptionally long 3′UTRs, with selenocysteine codons that are interpreted as nonsense codons in the absence of selenocysteine or undergoing programmed ribosomal frameshifting. Although in the literature these TCs are also often called PTCs, strictly speaking they are not “premature”, but normal termination codons (NTCs).

While most NMD targets are mRNAs, some small nucleolar (sno)RNAs and long non-coding (lnc)RNAs also appear to be regulated by NMD. These snoRNAs and lncRNAs are probably “bogus mRNAs” in that they or their precursors are physically associated with translating ribosomes. SnoRNAs are often contained in introns or alternatively spliced exons of mRNAs or lncRNAs. Processing of these precursor RNAs releases the snoRNAs, thereby creating PTCs in the “ORFs” of the spliced products, which are subsequently translated and degraded by NMD [23,24,25]. Some translated lncRNAs contain very short ORFs that encode functional microproteins or peptides [26] and therefore may be similar to regular NMD target mRNAs.

The heterogeneity of RNA targets is mirrored by the variety of attempts to explain the mode of action of NMD. Conceptually, NMD can be subdivided into a translation termination phase and an RNA decay phase. Roughly 15 years ago, two paradigmatic models, the “downstream marker” or “SURF” model [27] and the “*faux* 3′UTR model” [28] were shaped along these cornerstones to explain NMD of target mRNAs with spliced and unspliced 3′UTRs, respectively, and have inspired a wealth of research and insight, thus enabling constant refinement and variation of these two concepts.

Both models coalesce regarding the necessity of terminating ribosomes and a set of conserved NMD effectors. In Mammalia, these include the up-frameshift (UPF) factors UPF1, UPF2, and UPF3 and the suppressor with morphogenetic effect on genitalia (SMG) proteins (SMG1, SMG5, SMG6, SMG7, SMG8, and SMG9) for NMD to occur (see below). According to the downstream marker model, an aberrant termination complex consisting of the terminating ribosome, the UPF1 kinase complex *S*MG1-8-9 (SMG1c), the central NMD effector UPF1, and the eukaryotic release factors e*RF*1 and e*RF*3, forms at a PTC. This so-called SURF complex [27] is thought to delay translation termination and to sense the presence of an aberrant messenger ribonucleoprotein particle (mRNP) configuration on the 3′UTR. In mammalian cells such an aberrant mRNP constellation is represented by an exon junction complex (EJC) downstream of the PTC. EJCs are deposited during splicing at ~24 nucleotides (nt) upstream of exon-exon junctions [29,30]. EJCs serve as binding platforms for other proteins and have functions in NMD, in nuclear export of mRNA, and in enhancing translation (reviewed in this issue by Schlautmann & Gehring [31]). Most NTCs are located within last exons, and therefore most 3′UTRs do not harbor EJCs. The terminating ribosome and the EJC are thought to be bridged by simultaneous interaction of UPF2 with UPF1 at the termination site and with EJC-bound UPF3B [32], leading to the formation of a decay inducing complex (DECID) [27]. Assembly of the as yet hypothetical DECID probably involves additional proteins, such as neuroblastoma-amplified gene/neuroblastoma-amplified sequence (NAG/NBAS) [33], interactor of little elongator complex ELL subunit 1(ICE1) [34], serine/arginine-rich splicing factor 1 (SRSF1) [35], and the RNA helicases DEAH box protein 34 (DHX34) [36] and moloney leukemia virus 10 protein (MOV10) [37]. DECID formation is thought to trigger UPF1 phosphorylation, remodeling of the 3′ mRNP, recruitment of mRNA decay enzymes and dissociation of the release factors [27,38].

Similarly, the “*faux* 3′UTR” model also posits that translation termination at a PTC—in contrast to termination at an NTC—is slow and inefficient due an aberrant 3′UTR mRNP configuration [28,39]. However, in contrast to the SURF model the aberrant 3′UTR is signified by the lack of at least one termination promoting factor. Such a factor is Poly(A) binding protein (PABP) whose termination enhancing interaction with eRF3 may be disabled on unusually long 3′UTRs. Accordingly, the “*faux* 3′UTR” model is favored to explain NMD of RNAs with long 3′UTRs. NMD in this model is thought to be triggered either by the failure to recycle terminating ribosomes or by the recruitment of UPF1 to the aberrant termination site or its environment. Although originally derived from findings in yeast, the “*faux* 3′UTR” model has been successively expanded to accommodate EJC-independent metazoan NMD of targets with long 3′UTRs [40,41,42,43,44,45].

Central to both models is UPF1, a 123 kDA ATP-dependent 5′–3′ RNA/DNA-helicase and prototype member of the SF1 superfamily (also called UPF-like helicases [46]). UPF1 consists of an unstructured N-terminal region, a cysteine–histidine-rich (CH) zinc-knuckle domain, a helicase region and a carboxy-terminal serine and glutamine rich (SQ) domain. The SQ domain and the N-terminal region contain many S/T–Q motifs (Figure 1). The helicase domain consists of two RecA-like domains [47,48]. A pocket-like structure between RecA1 and RecA2 binds ATP [49,50,51,52]. The UPF1 helicase domain contains two additional regulatory subdomains 1B and 1C that effect UPF1′s high processivity and exert regulatory functions [46,47,52,53]. Alternative splicing results in two UPF1 isoforms (UPF1_1_ and UPF1_2_) that differ from each other exclusively by an 11-amino-acid insertion within subdomain B which confers enhanced RNA binding and catalytic activity to the UPF1_1_ variant [46]. The CH domain is an important modulator of UPF1 function and a hotspot of interaction with other proteins ([54] reviewed in [7]) (Figure 1). In UPF1′s inactive configuration, the CH domain together with the C-terminus sterically inhibits UPF1′s helicase and ATPase functions both of which are essential for NMD [32,47,49,55]. Binding of UPF2 to the CH domain and UPF1 phosphorylation by SMG1 are thought to induce a conformational rearrangement. This conformational change activates the ATPase and highly processive helicase activity of UPF1, thus allowing remodeling of the 3′ UTR RNP of NMD targets [47,49,53,56,57,58,59]. In addition to UPF2, UPF3B is also thought to contribute to the activation of UPF1 functions [32,44]. UPF3 comes in two variants: UPF3B (also known as UPF3X [60]) and UPF3A (also known as UPF3). UPF3A has long been considered as a backup molecule that can substitute for UPF3B in NMD albeit with a much lower efficiency [61,62]. More recently it has been suggested that UPF3A can act in certain situations as a UPF3B antagonist and suppressor of NMD by competing with UPF3B for UPF2 binding [63].

In yeast, all three UPF proteins are essential for NMD, thus composing a linear NMD pathway. By contrast, in higher eukaryotes, UPF2-independent, UPF3B-independent and EJC-independent NMD branches have been described (see below).

## 2. The Termination Phase of NMD

Normal translation termination ensues when the A-site of an elongating ribosome is occupied by a stop codon. During translation termination, eRF1 in complex with eRF3A and GTP binds to the stop codon, inducing a conformational rearrangement of the ribosome [64,65]. GTP hydrolysis by eRF3A leads to accommodation of the GGQ motif of eRF1 in the peptidyl transferase center of the large ribosomal subunit, resulting in rapid peptide release. After GTP hydrolysis and peptide release eRF3A dissociates from the post-termination ribosomal complex (postTC) and is replaced by the ABC-type ATPase ABCE1, which stimulates the splitting of the ribosomal subunits (reviewed in [66,67]).

The *modus operandi* by which translation termination at a PTC is distinguished from termination at a normal termination codon (NTC) is still poorly understood. According to a unified model, trying to integrate all available data in yeast and higher eukaryotes, the NMD effector UPF1 interacts with elongating ribosomes. This principally weak association of UPF1 with ribosomes is stabilized by UPF2 and UPF3. The formation of the UPF1,2,3-ribosome complex delays translation termination, and enables the NMD factor-terminating ribosome-complex to sense the presence of an aberrant mRNP on the extended 3′UTR. After peptide release, UPF2 and UPF3 promote ATP hydrolysis by UPF1 to fuel the dissociation of post-terminating ribosomal complexes. UPF1, possibly still bound to the 40S ribosomal subunit, then recruits mRNA decay enzymes to initiate mRNA degradation [28,44,68,69,70,71].

All concepts of a crosstalk between translation termination and the NMD machinery are largely based on genetic data in cells, analyses of protein–protein and protein–RNA interactions in vivo and on structural studies of complexes consisting of NMD factors and EJC proteins. Because no adequate in vivo termination assay is available, it has been challenging to address key molecular mechanisms that underlie the interaction of the NMD machinery with the translation termination apparatus. Moreover, the questions of how NMD recognizes its targets, how and when NMD factors are recruited to the target RNA, and how the dynamic interaction between RNA, terminating ribosome, and NMD effectors is orchestrated remain unsolved.

## 3. The Decay Phase of NMD

According to most traditional and emerging alternative NMD models, UPF1 directly or indirectly recruits RNA decay enzymes after translation termination and SMG1-mediated UPF1 phosphorylation at clusters of N- and C-terminal S/T–Q sites [72]. The PI3-kinase-like kinase SMG1 forms a complex (SMG1c) with the accessory factors SMG8 and SMG9 who regulate the activity and substrate specificity of SMG1c [38,73,74,75,76]. UPF2 and UPF3B are thought to stimulate the phosphorylation of UPF1 [27,44]. Besides providing a binding platform for RNA decay enzymes in the vicinity of the PTC or on the 3′UTR of NMD targets [77,78], several additional functions of UPF1 phosphorylation have been suggested. These include stimulation of UPF1′s release from the eRFs [27,72], repression of further translation initiation [79], activation of UPF1′s RNP remodeling function [27,44], and discrimination between NMD targets and non-targets [77,78]. No individual phosphorylation site of UPF1 is essential or sufficient to trigger NMD. Instead, progressive phosphorylation of S/T–Q sites at both the N-terminal region and the C-terminal SQ domain cumulatively contribute to the activation of partly redundant decay pathways [78,80,81,82,83]. Whether the phosphorylation of different S/T–Q sites and the resulting recruitment of decay enzymes or adaptors to degradation machines is sequential or otherwise coordinated remains to be determined.

UPF1 phosphorylation plays a role in recruiting SMG6 and SMG5/7 to NMD targets [72,82,84]. SMG5, SMG6, and SMG7 are directly or indirectly involved in target mRNA degradation but also stimulate UPF1 dephosphorylation by protein phosphatase 2A (PP2A) [85,86]. Phosphorylation at the N-terminal T28 site and the C-terminal S1096 and S1116 are crucial for binding of the endonuclease SMG6 and for SMG5 and/or SMG7 binding, respectively [72,82]. However, phosphorylation-independent interactions between UPF1 and SMG6 have also been reported [81,82]. According to several in vivo studies SMG6 initiates the decay phase of NMD by endonucleolytically cleaving NMD substrates directly at and/or in the vicinity of the PTC via its C-terminal PIN domain [24,87,88,89,90,91]. It is unknown whether transcript cleavage occurs before or after recycling of the terminating ribosome (reviewed in [92,93,94]). While also NMD substrates with long 3′UTRs are targeted by SMG6, endonucleolytic cleavage is considerably stimulated by the presence of EJCs [89]. Although it contains an EJC-binding domain (EBD), direct interactions between SMG6 and the EJC or with UPF2 are not necessary for endocleavage [89]. Instead, recent evidence suggests that CASC3 stimulates SMG6-dependent NMD and provides a link between the EJC and the NMD machinery [95]. UPF2 stimulates cleavage of targets with long 3′UTRs but seems to be dispensable for EJC-dependent NMD [89]. On targets with *bona fide* PTCs (i.e., on erroneous transcripts) the decay phase of NMD is constitutively initiated by SMG6-dependent endocleavage whereas NMD substrates with uORFs or long 3′UTRs are targeted both by SMG6 and by exonucleases recruited by SMG7 [88]. The SMG5/7 heterodimer recruits the CCR4/NOT complex via interaction with its catalytic subunit CNOT8/POP2 [96] and CCR4/NOT in turn recruits the decapping complex DCP1/DCP2 and the 5′-3′ exonuclease XRN1. UPF1 can also directly interact with the decapping factors DCP1 and PNRC2 independently of CCR4/NOT and/or SMG5/7. Direct recruitment of decapping factors by UPF1 may initiate decay of NMD substrates in yeast [54]. However, whether the interaction with PNRC2 plays a role in NMD is a matter of debate [97,98,99,100]. The decapped and deadenylated RNA body and the SMG6-induced decay fragments are subsequently exonucleolytically degraded from the 5′end by XRN1 and from the 3′ end by the exosome. Some phospho–UPF1- and ribosome-bound NMD decay intermediates are 3′ oligouridylated by the terminal uridylyl transferases TUT4 and TUT7 and degraded by the exonuclease DIS3L2 or by the exosome [101,102]. The 3‘ tagging of NMD targets has also been implicated in ribosome release from NMD targets [103].

While the marker model and the *faux* 3′UTR model have been prevailing for many years now, the number of inconsistent new findings have been mounting recently, leading to ever more frequent calls for and suggestions of alternative models [6,68,104,105]. In the following sections, we will discuss the findings that do not comply with traditional NMD models. Furthermore, we will critically examine the necessity of individual factors in NMD and other RNA decay pathways that also use UPF1 as central effector.

## 4. NMD Occurs on Transcripts with Nuclear or Cytoplasmic Cap-Binding Factors and in Each Round of Translation

A long-standing hypothesis posits that only NMD targets that are still bound to the nuclear cap-binding complex (CBC) are recognized by the NMD machinery and are rapidly degraded in the first (pioneer) round of translation which strips away all EJCs from the coding region (reviewed in [106]). As soon as the nuclear CBC is exchanged for the cytoplasmic cap-binding factor eIF4E, NMD substrates are thought to be immune to NMD. However, it has meanwhile been shown that NMD is also induced on eIF4E-bound mRNAs, so that the cap-binding factors play no role in the discrimination between NMD-sensitive and–insensitive transcripts in human [107,108,109] and yeast cells [110,111]. Consequently, these studies have also demonstrated that NMD can—at least in principle—ensue in any round of translation. This notion is strongly supported by a very recent report, which, using single-molecule imaging, provides multiple lines of evidence that each terminating ribosome has an equal probability of inducing NMD [112] thus confirming an earlier hypothesis [68]. Since NMD appears to be a fast process, this does not exclude that many targets are degraded during the first round of translation while they are still bound to the nuclear CBC.

## 5. Human UPF1 Does Not Directly Interact with the Release Factors

Central to conventional NMD models is the notion that the interaction of UPF1 with the termination factors eRF1 and eRF3 somehow brandmarks a termination event as aberrant. According to a thought-provoking hypothesis, a main (if not the only) function of Upf1p in yeast is the disassembly of prematurely terminating ribosomes, a role that can alternatively be supplemented or substituted in this organism by Upf2p or Upf3p [69,104,105,113].

Early studies in yeast showed that in pulldown assays, purified recombinant Upf1p binds to both eRF1 and eRF3 and that Upf2p and Upf3p also interact with eRF3 and compete with each other and with eRF1 for eRF3 binding [114,115]. Multiple lines of evidence including co-immunoprecipitation (co-IP) experiments [27,38,43,44], interaction studies using semi-purified proteins [43], as well as the finding that UPF1 binds to ribosomes in both yeast and human systems [116,117,118] appeared to validate this hypothesis. However, a recent study complementing such co-IP experiments with in vitro assays demonstrated that the eRF-UPF1 interaction is likely indirect. Enzymatically functional recombinant human UPF1 did not bind to either eRF1 or eRF3A. Furthermore, in a fully reconstituted in vitro translation system, neither UPF1 per se nor its biochemical functions such as ATP binding, ATP hydrolysis, or phosphorylation played a discernible role in early or late phases of translation termination [117,119,120]. Finally, directed hydroxyl radical probing revealed that while Upf1p binds the 80S ribosomal E site rRNA, it does not affect elongation, termination, or recycling in a fungal in vitro reconstituted system [118]. Taken together, this evidence indicates that human UPF1—even if mediating some indirect interaction with the eRFs—remains inactive during the termination phase, and instead likely exerts its indispensable role in NMD during the decay phase.

## 6. If Not by the eRFs, How Is UPF1 Recruited to NMD-Target Transcripts?

The concept of a SURF complex specifically forming at prematurely terminating ribosomes perseveres. However, it has been challenged not only by the finding that UPF1 does not directly interact with the release factors, but also by the discovery that UPF1 binds to some extent along the entire length of (possibly) all transcripts. UPF1 bound to the coding region is displaced by translating ribosomes. Therefore, UPF1 is mainly found on 3′UTRs in a length-dependent manner suggesting a role for UPF1 in 3‘UTR length sensing [121]. Steady state levels of 3′UTR-bound UPF1 are a result of continuous binding and dissociation [37,121,122,123]. Whether UPF1 enrichment on PTC-containing transcripts (and specifically on their 3′UTRs) contributes to NMD target identification is a matter of ongoing debate [45,77,122,124,125,126,127]. While UPF1 can be found on all 3′UTRs, its ATPase activity has been reported to promote UPF1 release preferentially from non-target mRNAs [125]. ATP binding reduces UPF1′s affinity for RNA and activates its helicase activity [46,52,128]. UPF1 mutants unable to bind or hydrolyze ATP display enhanced RNA binding and are locked on target and non-target mRNAs while hyperactive ATPase mutants lose their RNA binding competence. ATPase-deficient UPF1 is hyperphosphorylated and stabilized on the mRNA by SMG5/7 [52,77,125]. Preferential UPF1 ATPase-dependent release from non-target RNAs has recently been reported to be promoted by PTBP1 and possibly also by hnRNP L and PABP that were all previously found to shield mRNAs from NMD [129,130,131]. Furthermore, more UPF1 molecules accumulate on 3′UTRs of target than non-target mRNAs in a translation-dependent manner which is further enhanced by the presence of a 3′UTR EJC [123]. However, while incremental hyperphosphorylation of UPF1 bound to NMD targets and preferential ATPase-dependent release of UPF1 from non-targets may be a means to differentiate between these two kinds of transcripts, these processes alone are not sufficient to initiate or inhibit mRNA decay [78,79,125,132]. Finally, a recent report provided evidence that in *Drosophila melanogaster* (*D. melanogaster*), UPF1 is co-transcriptionally loaded onto mRNAs and is essential for their nuclear export, confirming that no NMD-specific recruitment to target RNAs is required [133]. It remains to be determined whether this also applies to mammals.

## 7. UPF1–PABP Competition and Distance between the TC and PABP

The distance between the TC and PABP has been found in many studies to influence the half-life of NMD targets and has served as an explanation for the NMD-sensitivity of transcripts with long 3′UTRs. Tethering PABP to the vicinity of a PTC renders the transcript immune to NMD in both yeast and mammalian cells [28,41,42,43,44,45]. In a reconstituted in vitro translation system PABP enhances translation termination by direct interaction with eRF3A [70] via its C-terminus. Based on these findings it has been hypothesized that due to the longer-than-normal spacing between the terminating ribosome and the Poly(A) tail, UPF1 instead of PABP interacts with the eRFs and consequently triggers decay. However, because NMD suppression by PABP is not mediated by its eRF3A binding domain, the UPF1–PABP competition scenario seems unlikely [70,134,135]. Furthermore, in yeast neither the Poly(A) tail nor Pab1p is required for discrimination between PTC-containing and normal mRNAs by NMD [136]. Instead, mRNA closed-loop formation via interaction between PABP and eIF4G/A provides a reasonable explanation for both the termination promoting and NMD-suppressing functions of PABP [6,137,138,139]. Alternatively, instead of antagonizing NMD by stimulation of termination, PABP may protect mRNAs from NMD by promoting dissociation of UPF1 from 3′UTRs [129].

Moreover, 3′UTR length is no strict NMD determinant. Due to three-dimensional folding induced by RNP formation or intramolecular interactions the distance between the TC and Poly(A)-bound PABP may not be all that long on some seemingly long 3′UTRs, making the “real” spacing between PABP and the TC unpredictable. For example, hiCLIP (RNA hybrid and individual-nucleotide resolution UV cross-linking and immunoprecipitation) has revealed that duplex formation can bridge sites that are as far away from each other as the stop codon from the 3′ end of a 3′UTR [140].

## 8. Is Termination at a PTC Really Slower than at an NTC?

Early on, a kinetic dimension involving UPF1′s ATPase function and an unfavorable termination context was hypothesized to underlie the discrimination between normal and premature termination [141]. Later studies seemed to confirm the idea of slow and inefficient translation termination and ribosome recycling at PTCs due to the absence of a termination promoting 3′UTR mRNP structure that probably includes PABP [39,41,44,70,71,134,142]. However, according to a recent report, in vitro translation of reporter mRNAs followed by toeprinting in human cell lysates failed to detect a difference in ribosomal occupancy at the termination codons of reporter mRNAs with or without NMD-inducing 3′UTRs and with or without a Poly(A) tail [143]. These findings were corroborated by ribosomal profiling of endogenous NMD-sensitive and insensitive transcripts in HeLa cells, which revealed that ribosome occupancy at termination codons is similar in both types of transcripts. Therefore, NMD activation seems to be unrelated to ribosome stalling at a PTC [143]. Although thus the question whether termination kinetics is relevant for bridging translation termination and NMD activation remains unsolved, the stop codon context may still help to define a termination event as improper. Such contextual features could include aberrant mRNP domains, as suggested by both traditional models. However, the nucleotide sequence around the TC per se [144] or its posttranscriptional modification may also contribute to the discrimination of an NTC from a PTC. For example, adenosine methylation is enriched near stop codons and within 3′ UTRs in both mouse and human mRNAs which may be relevant for stop codon recognition or marking the beginning of natural 3′UTRs [145,146,147]. To our knowledge the possible influence of posttranscriptional mRNA modifications has not yet been investigated in the NMD field.

## 9. UPF3B and ABCE1 Functions in Early and Late Translation Termination

Instead of UPF1, UPF3B appears to play the roles in early and late translation termination that have been previously ascribed to UPF1. Recombinant UPF3B forms a stable ternary complex with eRF1 and eRF3A; this interaction involves the eRF3A N-terminus and the middle domain of UPF3B. When delayed translation termination at a PTC is experimentally simulated by complementing a fully reconstituted translation termination system with limiting amounts of eRFs, UPF3B delays termination and impairs peptide release. Furthermore, UPF3B dissolves ribosomal post-termination complexes (postTCs) after GTP hydrolysis by eRF3A and after peptidyl-tRNA hydrolysis by eRF1. Both functions are unaffected by UPF1, but reversed by UPF2 which has no effect on translation termination on its own [117,119,120]. Since UPF3B is not an ATPase, its activity is reminiscent of the energy-free activity of initiation factors that—like UPF3B—can recycle post-TCs at (physiologically) low Mg^2+^ concentrations [148]. Importantly, although no function was assigned to UPF2 in this reconstituted system, UPF2 has also been reported to directly interact with eRF3A [149]. Since UPF3B is involved in NMD activation on only a subset of NMD targets it is unlikely to have a general function in ribosome release at PTCs. Instead, in a recent pre-publication report, the ribosome recycling factor ABCE1 plays a critical role in resolving stalled ribosomes on a subgroup of NMD target mRNAs [150]. Future studies will determine whether UPF3B’s and ABCE1′s functions in ribosome release at PTCs are mutually exclusive or redundant.

## 10. Are There Any Indispensable *Trans*-Acting Factors or *Cis*-Acting Features in NMD?

Because NMD in higher eukaryotes appears to be a ramified pathway using partly redundant effectors to target RNA substrates with often obscure features, we will explore in the following if any of these factors or features are in the strict sense either sufficient or indispensable to elicit NMD.

### 10.1. UPF2, UPF3B, SMG1, and the EJC

Apart from UPF1, a variety of accessory factors play a role in both EJC-dependent and EJC-independent NMD. In yeast the NMD core factors Upf1p, Upf2p, and Upf3p are equally indispensable. However, the situation in metazoans has turned out to be far more complex. Although UPF2 has been ascribed several functions in modulating UPF1 activities including UPF1 phosphorylation, ATPase and helicase activities, it seems to function mainly in EJC-independent NMD and to be dispensable for EJC-dependent NMD [89,151,152]. Likewise, a UPF3B-independent NMD branch was described in human and mouse cells [62,153]. Both pathways are partially redundant in EJC-independent NMD, depend on UPF1 and SMG1 [154], and appear to have important functions in neural cell and brain metabolism, in embryonal development and in spermatogenesis [155,156,157].

To this day, NMD-related publications state that UPF2 bridges the terminating ribosome and the EJC by interacting both with UPF1 at the termination site and with EJC-bound UPF3B [32]. However, the foundations of this hypothesis were shaken some time ago. Since UPF2 seems to be dispensable for EJC-dependent decay [88,89,158], since UPF3B can bind to UPF1 without UPF2 [117], since the interaction between the EJC and UPF2 (and SMG6) are not required for endocleavage [89] and since both UPF2 and UPF3B can interact with eRFs (or the SURF complex) independently of each other and the EJC [117,149], this bridging function is either non-existent or plays a subordinate role in NMD, thus demolishing another important corner stone of the 3′ marker model.

Hence, neither UPF2 nor UPF3B are essential NMD factors in the strict sense that NMD cannot function without them. Because the original SURF/DECID concept cannot be held up, what may be the function(s) of UPF2 and UPF3B in NMD? Both have been reported to assist in UPF1 phosphorylation in vivo [27,44] but in vitro UPF2 only slightly stimulates UPF1 phosphorylation. Conversely, UPF3B alone moderately and together with UPF2 considerably impairs UPF1 phosphorylation by SMG1-8-9 [117]. Instead, UPF2 seems to facilitate the release of UPF1 from the SMG1-8-9 complex after UPF1 phosphorylation [74]. UPF2 has also been implicated in activation of UPF1′s helicase and ATPase functions by unblocking the intramolecular inhibition of UPF1 by its CH domain (see above). While UPF1 phosphorylation may be promoted by DHX34 in addition to or instead of UPF2 and UPF3B [83], it is unclear how UPF1′s important functions in mRNP formation and remodeling is activated in the absence of UPF2.

While UPF1 phosphorylation seems to be essential in mammalian NMD, it is dispensable in several species including *Saccharomyces cerevisiae (S. cerevisiae)* and *Arabidopsis thaliana* either due a lack of UPF1 S/T–Q-sites or loss of SMG1 and its accessory factors SMG8 and SMG9 (reviewed in [12]).

In *S. cerevisiae*, only a minority of pre-mRNAs undergo splicing and thus yeast NMD has traditionally been considered to be EJC-independent. In *D. melanogaster*, EJCs are deposited at splice junctions, but NMD of only few substrates has been reported to be EJC-dependent in this organism. The specification of termination events as aberrant by the presence of at least one EJC on a splice junction downstream of the termination codon was first described in mammalian cells more than 20 years ago [159,160,161]. However, since the discovery of EJC-independent NMD in mammals it is generally acknowledged that in these organisms, too, EJCs are not strictly essential for NMD, although their presence dramatically enhances its efficiency. It is not unequivocally clear which EJC core factors are required for EJC-dependent NMD. Several peripheral EJC proteins including UPF3B, CASC3, RNPS1, and ICE1 [34,95,162] appear to function in redundant or exclusive NMD branches and/or to recruit further NMD-relevant proteins to the 3′UTR. Therefore, the main function of the EJC in NMD may be to increase the local concentration of such proteins [8]. The NMD-promoting function of the EJC has been further challenged by the discovery that in *Schizosaccharomyces pombe* and *S. cerevisiae*, splicing both upstream and downstream of PTCs enhances NMD, but is independent of EJC components [163,164]. Similarly, according to a recent report, the number of introns determine NMD efficiency in mammalian cells regardless of their position upstream or downstream of a PTC [112].

### 10.2. Decay-Inducing Factors

The situation is similarly ambiguous regarding factors that are directly or indirectly involved in the degradation of NMD substrates. Of two studies in mammalian cells with partly conflicting results, one concluded that the extent to which SMG6 and/or SMG7 direct NMD substrates to at least partly non-overlapping decay pathways is determined by the mRNA architecture [88]. The other found that SMG6 and SMG7 act redundantly on largely the same target RNAs and define a homogenous pathway [23]. The fact that simultaneous depletion of SMG6 with SMG5 or SMG7 is more effective than individual depletion of these factors appears to indicate that the two pathways are only partly redundant. Moreover, UPF1 can recruit decapping factors independently of SMG5/7 (reviewed in [6]). SMG7 is not required for NMD in *D. melanogaster* and SMG6 depletion does not fully abrogate NMD in this organism [165,166]. Taken together, these findings indicate that SMG6, SMG5, SMG7, and the SMG5/7 heterodimer are neither fully sufficient nor strictly necessary for NMD.

### 10.3. Translation Termination

Whether UPF1 depletion or deletion in mammals and deletion of any of the UPF factors in yeast, influences translation termination fidelity is controversial [44,167,168,169]. However, in contrast to two other translation-dependent RNA surveillance mechanisms, non-stop decay (NSD) and No-Go Decay (NGD), NMD uses the normal translation termination machinery (reviewed in [92,94,170]) and at least some NMD targets probably also use the normal ribosome recycling machinery [150]. Whether others make use of UPF3B for the disassembly of ribosomes terminating at a PTC, as in vitro finding suggest [117] has yet to be confirmed in vivo. Therefore, while the necessity of a translation termination event is indisputable, whether and how the nature of this termination event differs from normal termination remains obscure.

### 10.4. Target RNA Features

Exon–exon junctions, regardless of whether they are bound by an EJC and possibly regardless of their position relative to the termination codon, strongly enhance NMD in several (but not all) organisms. As discussed above, long 3′UTRs can be NMD determinants, but whether the cell “interprets” a 3′UTR as long depends on its three-dimensional structure. 3′UTR-bound UPF1 has been reported to be enriched on G/GC-rich sequence clusters [124,171] which may indicate a binding preference, but may also reflect slower helicase processivity of UPF1 due the fact that such stretches often form secondary structures. Likewise, some, but not all uORFs direct mRNAs to NMD. Decay of uORF containing transcripts can be prevented by reinitiation promoting features such as downstream canonical or non-canonical translation reinitiation sites or strong secondary structures between the uORF and the main ORF [172,173,174,175].

Strictly speaking, when considering *cis*-acting RNA elements, only a termination codon is required to trigger NMD; however, with the exception of true PTCs, most termination codons of endogenous NMD targets reside at the end of regular ORFs. When considering *cis*-acting RNA sequence elements, NMD-promoting RNA features seem more difficult to identify than NMD-restricting features. In terms of protective elements, potential NMD substrates with long 3′UTRs can be shielded from NMD by RNA-binding proteins (RBPs) that bind to *cis*-acting sequence elements [176,177] in the vicinity of stop codons and may at least in some cases compete with UPF1 for RNA binding (see below; reviewed in [8,9]. Such proteins include the polypyrimidine tract binding protein 1 (PTBP1) [131] and hnRNP L [130] which preferentially bind to CU-rich and CA-rich sequences, respectively.

Taken together, like with *trans*-acting NMD factors, there are no unequivocal *cis*-acting NMD-triggering RNA features and thus it remains hitherto impossible to reliably predict if a given RNA is an NMD target [6].

### 10.5. UPF1

While all other protein factors are neither indispensable nor sufficient to trigger NMD, it is generally acknowledged that UPF1 is the core NMD player—or almost so. UPF1, although expressed, is not required for a NMD-type RNA decay pathway in *Trypanosoma brucei* [178], and deletion of *UPF1* and *UPF2* does not completely suppress NMD in *S. pombe* [163,179]. Aside from these rather exotic examples and judged by general biochemical and cell biologic standards, UPF1 *per* se as well as its phosphorylation and its helicase and ATPase functions appear to be necessary for NMD in almost all eukaryotic organisms.

In summary, while UPF2, UPF3B (together with its regulator UPF3A), SMG5-7, and the EJC (including some of its peripheral proteins such as CASC3 and RNPS1), help to direct mRNAs with a variety of architectural features to the NMD-type degradation pathway, none of these proteins or RNA features is indispensable or sufficient. The only absolute necessity is a translation termination event on an mRNA that either recruits and activates UPF1 subsequent to termination or induces activation of pre-bound UPF1. However, this is also true for a number of other UPF1-mediated mRNA decay (UMD) pathways (Table 1).

## 11. Staufen-Mediated mRNA Decay (SMD)

SMD degrades mRNAs with Staufen (STAU)1/2-binding sites in their 3′UTR. STAU-binding sites (SBS) are double-stranded (ds) RNA structures usually generated by intramolecular base-pairing [140]. In rare cases, STAU-binding sites can also arise from intermolecular base pairing between *Alu* short interspersed nuclear elements (SINE) within the 3′UTR of mRNAs and partially complementary SINEs within lncRNAs [180,181]. The length of 90% of RNA duplexes bound by STAU1 range from 5 to 14 nt lacking a precise sequence specificity; however, duplexes can enclose extraordinarily long loop structures of several hundred nucleotides [140]. In Mammalia, two *D. melanogaster* Staufen paralogs, STAU1 and STAU2, are expressed that regulate the expression of partly overlapping sets of target mRNAs [182]. Like NMD, SMD depends on translation termination and requires the recruitment of UPF1 as well as its phosphorylation (probably) by SMG1 [183]. The STAU proteins were originally thought to directly recruit UPF1 to substrate mRNAs and to activate UPF1′s ATPase and helicase functions in a manner analogous to UPF2 function in NMD [184]. However, a recent report demonstrated that STAU1 exhibits only weak direct interactions with UPF1 and is unable to directly stimulate UPF’s catalytic functions in vitro. Instead, UPF2 binds to a STAU1 dimer, recruits UPF1 to the STAU1 mRNP, and activates UPF1′s catalytic functions by binding to the CH domain, thus releasing the helicase and ATPase domains. Furthermore, SMD is suppressed in vivo upon depletion of UPF2 [185]. These findings may also help to better understand why NMD and SMD compete with each other in cells [184]. NMD–SMD competition was thought to reflect a competition of UPF2 and the STAU proteins for binding to UPF1′s CH domain. However, it seems that UPF2 enhances rather than prevents binding of STAU1 to UPF1. STAU1 contacts the MIF4G domain of UPF2 which is also where UPF3 binds to UPF2. However, the binding sites are probably not identical, simultaneous binding of STAU1 and UPF3 may be impossible, thus offering an alternative explanation for the competition of SMD and NMD [185].

## 12. Histone mRNA Decay

Replication-dependent histone mRNAs lack a poly(A) tail and instead harbor a 26 nt stem loop (SL) starting between 20 and 75 nt 3′ to the TC [186]. The stem loop is specifically bound by stem loop-binding protein (SLBP). SLBP is required for all steps of histone metabolism. Both, the stem loop and SLPB mediate rapid degradation of histone mRNAs when DNA replication is completed at the end of the S phase or inhibited under cellular stress. After (possibly inefficient) translation termination, UPF1 is enriched in the region between the TC and the SL [187,188] and is thought to be activated by SMG1-mediated phosphorylation [189]. Alternatively, UPF1 phosphorylation by the PI3 K-like kinases ATR and DNA–PK have also been considered as a driver of histone mRNA decay (HMD) [190,191]. HMD requires 3′ end uridylation by TUT7 prior to decapping and exonucleolytic degradation [189,192]. Alternatively, it has been suggested that the complex between SLBP and phosphorylated UPF1 recruits SMG5 and the decapping enzyme PNRC2 [191]. It is as yet unclear whether SLBP’s binding not to the CH, but to the helicase domain of UPF1 can directly activate UPF1′s helicase and ATPase functions or whether, alternatively, UPF1 phosphorylation is sufficient to perform this task [186,191].

## 13. Regnase-1 Mediated mRNA Decay

The RNA-binding proteins regulatory RNase I (regnase-1, also known as ZC3H12A [193]) and roquin regulate an overlapping set of inflammation-related mRNAs. While they bind on their target mRNAs in a mutually exclusive manner to the same simple 3′UTR stem loop (SL) structure with a bulge consisting of the sequence UAU (or pyrimidine-purine-pyrimidine), their action is separated in space and time. Regnase-1 degrades mRNAs including *IL6* and *NFKBZ* transcripts during the early phase of inflammation and colocalizes with cytoplasmic and rough endoplasmic reticulum polysomes, indicating a function during active translation. In contrast, roquin destabilizes translationally inactive mRNAs at later stages of inflammation and localizes to *P*-bodies or stress granules [194].

Regnase-1 contains a PIN domain responsible for its endonucleolytic activity. *In vitro,* regnase-1 binds to the *IL6* SL, but is unable to cleave it, while addition of recombinant WT UPF1, but not of ATPase/helicase-deficient UPF1 induces rapid cleavage of the RNA substrate. UPF1 helicase activity unwinds the SL thus enabling subsequent cleavage by regnase-1 even in the absence of UPF1. Thus, a structural change in the target RNA mediated by UPF1 acts as a molecular switch to activate the RNase activity of regnase-1 and the degradation of target RNAs [195]. Similarly, translation and UPF1-binding are dispensable for regnase-1 binding to the target SL in vivo, but are required for its function in destabilizing translationally active mRNAs [195]. Upon translation termination ≥ 20 nt upstream of the SL, SL-bound regnase-1 recruits UPF. The intrinsically disordered regnase-1 linker region between amino acids 90 and 130 binds directly to the UPF1 RecA helicase domain. This interaction considerably enhances UPF1′s ATPase and helicase activity and is therefore thought to release the UPF1 CH domain from the helicase region in a manner analogous to UPF2 binding to the UPF1 CH domain during NMD. In this context it is important to note that UPF2 and UPF3B appear to be dispensable for regnase-1 mediated mRNA decay (RMD) [194]. Furthermore, this regnase-1-UPF1 interaction induces phosphorylation of UPF1 T28 by SMG1. T28 phosphorylation enables a second direct interaction between the N-terminal UPF1 region containing T28 and the regnase-1 PIN domain and stabilizes the interaction between UPF1 and regnase-1 in vitro and in vivo [195]. Because a UPF1 C-terminal deletion mutant binds even better to regnase-1 it is likely that the UPF1 C-terminus impairs both, the helicase function of UPF1 and regnase-1 binding and thus may contribute to the spatiotemporal control of RMD. Inhibition of SMG1 in bone marrow-derived dendritic cells (DC) prevents UPF1 phosphorylation at T28, resulting in increased expression of RMD targets and the encoded proinflammatory cytokines. This result indicates that SMG1 regulates cytokine expression and DC maturation in a regnase-1-dependent manner in vivo.

## 14. TRIM71-Mediated mRNA Decay (TRIM71-MD)

TRIM71 is a stem cell specific mRNA repressor protein [196] that plays an essential role in early embryogenesis and in carcinogenesis within a narrow spatio-temporal window. Besides an E3 ubiquitin ligase RING domain enabling autoubiquitylation and ubiquitylation of several other proteins [197], TRIM71 contains a six β-propeller NHL domain essential for its mRNA repressing function. Via this NHL domain TRIM71 directly interacts with a stem/3-nucleotide-loop motif in the *CDKN1A*/*p21* 3′UTR. TRIM71 binding to this TRIM71 responsive element (TRE) induces target mRNA degradation [198]. TRIM71-mediated degradation of endogenous *CDKN1A* mRNA and of a reporter mRNA furnished with the *CDKN1A* 3′UTR was impaired after knockdown of NMD factors SMG1, UPF1 and SMG7, but not of SMG6 in HEK293 T and HepG2 cells. Importantly, depletion of these NMD factors had no effect on *CDKN1A* mRNA expression in the absence of TRIM71. Moreover, TRIM71 co-immunoprecipitates with UPF1, SMG1, and SMG7 in an RNA-dependent manner, suggesting that they bind to a common RNA target. These findings demonstrate that TRIM71 cooperates with NMD factors to mediate degradation of *CDKN1A* mRNA degradation. Further work revealed that TRIM71 and UPF1 cooperatively regulate the expression of a number of known EJC-independent NMD targets (all of which contain a 3′UTR longer than 750 nt), but not of „canonical“ EJC-dependent NMD targets [198]. It remains to be investigated whether TRIM71 directly recruits UPF1 to the 3′UTR, whether TRIM71 and UPF1 bind the RNA cooperatively or independently or whether, alternatively, a weak direct TRIM71-UPF1 interaction is stabilized by RNA or—as in the case of SMD—by an as yet undiscovered protein (such as UPF2). Taken together TRIM71-MD may represent one of several (or many) alternative forms of EJC-independent long 3′UTR mediated NMD.

## 15. GC-Rich-3′UTR Mediated mRNA Decay (GC-rich 3′UTR-MD)

UPF1 binds to normal nascent mRNAs along both coding regions and UTRs and is enriched on 3′UTRs after translation-dependent displacement from the coding region [121,122,123,124,125]. A meta-analysis of datasets resulting from ribonucleoprotein (RNP) immunoprecipitation followed by high-throughput sequencing (RIP-seq) [171], phospho–UPF1 RNA footprinting [77], and cross-linking immunoprecipitation sequencing (CLIP-seq) [125,199] revealed that (phospho-) UPF1 is enriched on 3′UTR sequences with high GC content and binds preferentially to a CCUG[GA] [GA] [GA] motif. Messenger RNAs with several such 3′UTR motifs had short half-lives and were stabilized by depletion of UPF1, UPF2 and SMG1 [171]. UPF1 release from a reporter mRNA harboring GC-rich 3′UTR sequence motifs was delayed in comparison with a control reporter containing several G > A mutations, supporting an earlier hypothesis that UPF1 binds to possibly all mRNAs, but selectively dissociates in an ATPase dependent manner from non-target mRNAs [125]. Together with the finding that UPF1′s ATPase activity is weaker on G- and GC-rich sequences [128], and that GC-rich stretches tend to form secondary structures, this finding may indicate that translocating UPF1 stalls and subsequently becomes phosphorylated at GC-rich sequences on 3′UTRs. This phosphorylation may result in the recruitment of RNA degrading enzymes including SMG5-7. Hence, the GC-rich motifs may not be *bona fide* UPF1 binding sites, but instead cause an increase in UPF1 concentrations on 3′UTRs. However, whether UPF1′s ATPase and helicase functions are essential for GC-rich 3′UTR-MD has yet to be assessed.

## 16. UPF1/SMG7/miRNA-Mediated mRNA Decay (UPF1/SMG7/miRNA-MD)

The UPF1/SMG7/miRNA-mediated mRNA decay model partly builds on and extends the GC-rich 3′UTR model of EJC-independent long 3′UTR-mediated mRNA decay. Long 3′UTRs are more likely to contain microRNA (miRNA) recognition elements (MREs) that bind Argonaute (AGO)-loaded miRNAs. Motif enrichment analysis of 3′UTR sequences of UPF1-dependent, but EJC-independent mRNA decay targets based on RNA-seq data of UPF1 or control siRNA-treated cells revealed an extended CUG motif very similar to that of the GC-rich 3′UTR-MD [171]. A considerable proportion of CUG motifs were embedded in miRNA 7mer sites of abundant miRNA families and such miRNA target mRNAs were significantly upregulated in UPF1-depleted cells. This effect disappeared when the CUG motif was mutated and upon depletion or deletion of Dicer, suggesting a correlation between miRNA- and UPF1-mediated decay [200]. UPF1 and AGO binding sites have previously been shown to overlap [124]. Experiments using endogenous and reporter target mRNAs with WT or mutated MREs with embedded CUG motifs in UPF1 depleted cells revealed that UPF1 and AGO cooperate on targets of this UPF1-dependent decay pathway. Surprisingly, neither SMG5, SMG6, nor the helicase function of UPF1 appear to be necessary for UPF1-dependent miRNA-mediated mRNA decay. This indicates that UPF1 translocation on the 3′UTR is not required and mRNA decay is not initiated by endocleavage. (As a note of caution, it should be noted that the residual level of endogenous helicase-competent UPF1 after siRNA-mediated knockdown may be sufficient to perform the unwinding, translocation or RNP remodeling functions necessary for this type of UMD, even in the presence of an ectopically expressed helicase-deficient UPF1 mutant.) In contrast, SMG7-depletion upregulates the expression of a largely overlapping set of mRNAs containing MREs with CUG embedded motifs of the same miRNA families. SMG7 is recruited by UPF1 to a complex that also contains AGO and interacts with deadenylase complex components NOT1 and NOT3 in a SMG7 dependent manner. However, other than in the canonical miRNA pathway, GW182 (also known as TNRC6A/C) is not required for the interaction with the CCR4/NOT complex, indicating that UPF1/SMG7-dependent miRNA-mediated gene regulation may constitute an alternative miRNA targeting pathway [200]. UPF1/SMG7/miRNA-mediated mRNA decay has been dubbed UPF1-mediated decay (UMD) by its discoverers who posit that this decay pathway, rather than NMD, is the norm for EJC-independent degradation of mRNAs with long 3′UTRs and degrades between 40% and 50% of all error-free NMD substrates [200].

## 17. Structure-Mediated RNA Decay (SRD)

Structure-mediated RNA Decay (SRD) selectively degrades mRNAs and circular RNAs with highly structured regions in a UPF1-dependent, but (linear) sequence-independent manner [201,202]. Untranslated regions of mRNAs and untranslated RNAs are often highly structured [203]. SRD may overlap with GC-rich 3′UTR-MD and UPF1/SMG7/miRNA-MD because among other factors the structure of an RNA or RNA domain is determined by its GC content. Counterintuitively, mRNAs with structured regions often have shorter half-lives [204]. Evaluation of public datasets revealed that UPF1 binding to highly structured RNA regions depends on its helicase function. Most helicase-dependent UPF1-bound 3′UTRs are structured (highly structured 3′UTRs, HSUs) irrespective of their length and display UPF1-dependent shorter than average half-lives. However, SRD is independent of UPF2, UPF3A, UPF3B, SMG6 and of SMD and RMD effectors [201]. Meta-analyses of public proteomic and CLIP-seq data revealed that G3BP1 and G3BP2 (also known as Ras GTPase activating proteins) interact with UPF1, that they bind to many helicase-dependent UPF1 target 3′UTRs and that they preferentially bind to HSUs. Mutations of G3BP1′s RNA-binding domain impair G3BP1-dependent regulation of target RNAs with HSUs. As shown by depletion and co-immunoprecipitation experiments, UPF1 and G3BP1 cooperatively regulate HSU-RNAs independently of 3′UTR length. Surprisingly, the UPF1-G3RP1 interaction is indirect and mediated by highly base-paired RNA regions that are not identical with dsRNA structures bound by STAU proteins. UPF1 is necessary for this RNA decay pathway, but unable to direct its substrates to decay without G3BP1. Since specific binding of UPF1 to HSUs depends on its helicase activity it is conceivable that unwinding of dsRNA stretches by UPF1 facilitates G3BP1 binding in the vicinity of UPF1. However, since G3BP is itself a helicase that can unwind partial DNA or RNA duplexes it still remains to be determined whether HSU unwinding depends on UPF1 or if UPF1 and G3BP1 act cooperatively to accomplish this task. The nucleases acting in this pathway and whether SMG5/7 are involved has yet to be studied.

## 18. Tudor-Mediated miRNA Decay (TumiD)

Tudor-staphylococcal/micrococcal-like nuclease (TSN) is an evolutionarily conserved component of the RNA-induced silencing complex (RISC) [205] and degrades a subset of miRNAs that contain CA and/or UA dinucleotides at conserved positions [206]. This process, named Tudor-mediated miRNA decay (TumiD) degrades miRNAs that regulate mRNAs encoding proteins with a role in G1-to S-phase transition and in cancer cell invasion [206,207]. TSN and UPF1 directly interact in vitro and co-immunoprecipitate from human cell lysates. In presence of a crosslinker they form an RNA-independent complex with RISC components AGO2 and GW182 which may indicate that a preformed TSN-UPF1 complex transiently interacts with RISC bound target miRNAs [207]. Up to 50% of TSN-regulated mature miRNAs (but not pri- or pre-miRNAs) are also upregulated by UPF1 depletion, but insensitive to knockdown of UPF2, UPF3B, or STAU1, indicating that UPF1-function in TumiD is independent of its role in NMD and SMD. UPF1′s helicase function (but not its C-terminal phosphorylation) is essential for TumiD. Hence, recombinant UPF1 can unwind miRNA/target RNA duplexes and promote degradation of the miRNA partner in vitro. It also induces release of pre-formed 3XFLAG-AGO-miRNA complexes from an in vitro synthesized target 3′UTR [207]. This indicates that UPF1 helicase function makes target miRNAs accessible to the nuclease TSN and may also contribute to recycling RISC-bound miRNAs from their mRNA targets.

## 19. Glucocorticoid Receptor-Mediated mRNA Decay (GMD)

GMD is a very interesting addition to the UMD family that differs in some important aspects from most other UMDs because it is independent of translation and requires UPF1 recruitment to a region that on natural target mRNAs is often located in the 5′UTR [208,209]. Cytoplasmic glucocorticoid receptor (GR) binds to a GC-rich motif forming two contiguous loops on a short stem within the 5′UTRs of a subset of mRNAs, thereby regulating their turnover rates [209]. GR-binding sites experimentally inserted into 3′UTRs can also elicit GMD [208]. GR can bind to mRNAs in the absence of its ligand glucocorticoid (GC). Upon ligand binding RNA-bound GR recruits PNRC2, thus creating a binding platform for UPF1 and DCP1A and inducing rapid mRNA decay [208]. In contrast to NMD, SMD, HMD, and other UMDs, GMD still proceeds when translation is blocked [208,210], proving that translation (termination) is not indispensable for UPF1 activation. GMD requires UPF1′s ATP binding, ATP hydrolysis and helicase functions as well as UPF1 phosphorylation by the PI3 K-like kinase ATM; however, it is independent of SMG1 [208] and SMG5-7 [210]. Ligand binding induces the formation of a GR-PNRC2-UPF1-DCP1A complex (“GMD complex I”) that subsequently recruits YBX1 and HRSP12 resulting in the active “GMD complex II” [210]. HRSP12 is an endonuclease that may serve in GMD in a manner analogous to SMG6 in NMD. However, HRSP12 has recently been found to lack endoribonuclease activity in vitro and to serve only as an adaptor protein in the cleavage of m^6^A-containing RNAs [211].

## 20. A Growing List

More forms of UMD are to be expected. The newest addition to the collection named endoplasmic reticulum (ER) NMD is presented by a very recent pre-publication [212]. NBAS (neuroblastoma-amplified sequence) is a component of the Syntaxin 18 complex involved in Golgi-to-ER trafficking and was previously found to play a role in NMD of a subset of target mRNAs [33,213]. NBAS is a peripheral ER protein and recruits hypophosphorylated UPF1 to the ER membrane. ER–NMD specifically degrades mRNAs that are localized to and translated at the ER. Both NBAS and UPF1 are necessary for this pathway and interact with the translocon component SEC61β which localizes them to the translation site of ER-destined mRNAs encoding secretory or integral membrane proteins. UPF2 appears to play only a minor role in this UMD, if any. It remains to be determined if this pathway is, as the authors imply, a subtype of NMD that requires other NMD effectors such as UPF3B, SMG1, SMG6, or SMG5/7 or if it is one of the UMDs that have only UPF1 in common with NMD. Interestingly, ER NMD (or ER UMD) is in accordance with previous studies showing a co-regulation of NMD efficiency and unfolded protein response under cellular stress and opens a route to finetune ER homeostasis by differentially modulating ER–NMD efficiency [20,214,215,216,217].

## 21. Crosstalk between Decay Pathways

In the light of the UMDs elucidated above, it is obvious that there is a certain overlap between different types of UMDs, but also between UMDs and other degradation pathways. In addition to UPF1, some UMDs share other canonical NMD factors and/or depend on translation (termination) (Table 1). Most UMDs not only require UPF1 as such, but also UPF1′s phosphorylation and/or its ATPase and/or helicase functions. UPF1/SMG7/miRNA-MD and TumiD share components with the miRNA-mediated gene silencing pathways. Both HMD targets and some ribosome-bound NMD decay intermediates are oligouridylated by TUTases as a necessary step of their regulated degradation [101,102,189]. Oligouridylation is generally important for the regulated turnover of many RNAs, including miRNAs and mRNAs (reviewed in [218,219,220,221]).

A comprehensive study, monitoring endocleavage, deadenylation and decapping in several RNA decay pathways including AU-rich element (ARE)-mediated decay, miRNA-mediated decay, RMD and NMD [222] revealed that the *TNF-α* and *IL6* RMD substrate mRNAs contain both AREs and regnase-1-binding stem loops. Surprisingly, while the 3′UTRs of both substrates are subjected to endocleavage, the cleavage sites are outside the regnase-1 binding sites and endocleavage could not fully explain the rapid decay of both substrates. Therefore, the substrates are likely simultaneously targeted by RMD and other decay pathways. An independent study points in the same direction. UPF1 hyperphosphorylation is activated upon depletion of SMG6, SMG7, and XRN1, but also transiently increased in response to activation of the ARE-mediated decay pathway. This indicates that two independent pathways can act on the same substrate and compete for shared downstream decay effectors [78].

Furthermore, in flies and worms, nonstop decay (NSD), another translation-coupled mRNA surveillance pathway that degrades transcripts lacking a stop codon, but (in contrast to NMD) does not use canonical translation termination factors, intersects with NMD in the decay phase [223,224]. Studies on NSD in both organisms revealed that NMD decay intermediates, resulting from endocleavage at or upstream of a PTC are degraded by the NSD machinery. In flies, NMD 5′ intermediates resulting from endonucleolytic cleavage 5′ to the stop codon (probably between the penultimate ribosome and the terminating ribosome) are targeted by NSD [223]. Truncated Ribo-seq reads at stop codons obtained from ribosomal profiling experiments revealed that most endogenous NSD targets in *C. elegans* are probably derived from NMD decay intermediates [224]. These findings indicate that in these two organisms NMD generates NSD targets and that NSD and NMD share important components of their decay machinery. It remains to be determined if such a cooperation also exists in mammals.

## 22. Targets of More Than One UPF1-Dependent Degradation Pathway?

Several UMD target RNAs that were first identified as substrates of traditional NMD (branches) have later been assigned to other UMDs. This may have several reasons: (1) the pathways may have overlapping target populations, thus ensuring decay on several levels; (2) not all UMDs may be present or equally efficient in all cell types or tissues, so that some UPF1-dependent RNA decay substrates may be targeted by different UMDs in different cell types; (3) there is no sharp border between the UMDs; (4) targets may be “misannotated” due to overlooking features that qualify them for other pathways. This may be the case for at least some UMD substrates (Table 2). The PTGS2 mRNA, for example, is a representative target for both the UPF2-dependent NMD branch [151,214] and for RMD [195]. The PTGS2 transcript has a long 3′UTR, but also a uORF qualifying it for both EJC-dependent and -independent NMD, via the long 3′UTR for other UMDs and via a 3′UTR regnase-1-binding stem loop for RMD. Regnase-1 is in most cells expressed at low levels [225] and therefore PTGS2 transcript expression may normally be regulated by NMD. Upon proinflammatory stimuli or in cells and tissues involved in the immune response PTGS2 may be a target of RMD rather than NMD. Similarly, the PEA15 transcript has been assigned both to NMD [214] and UPF1/SMG7/miRNA-MD [200]. PEA15, like PTGS2 mRNA, contains both a long 3′UTR and a uORF and therefore can likely be degraded by both pathways, either simultaneously or exclusively, depending on cell type, physiological situations, timing or subcellular location.

## 23. To NMD or Not to NMD?

UMDs probably share sets of effectors and of substrates to a larger extent than currently known, raising the question of whether these decay pathways can be clearly distinguished from each other or rather are variants of a *danse macabre* choreographed by UPF1 (Figure 2).

NMD is certainly the best characterized of all UMDs. Nevertheless, as outlined above, except for the key regulator UPF1 none of its effectors or target RNA/RNP features are sufficient or indispensable. This includes the EJC, which is probably the strongest NMD supporting feature known to date. In contrast to longstanding convictions, even translation termination, although necessary, may not have any NMD-specific characteristics. The translation termination codons of error-free endogenous NMD substrates are normal termination codons which makes it difficult to speak of them as premature. On the other hand, if translation termination combined with certain 3′UTR mRNP features qualifies an mRNA as NMD substrate, it is hard to maintain a conceptual distinction between (for example) NMD and HMD or NMD and SMD. This conclusion is strengthened now that we know that for the latter not only UPF1, but also UPF2 is required.

This confusing situation is lately increasingly being recognized and several suggestions have been made to solve the problem, at least for what has hitherto been described as NMD. By analogy, other UMDs can likely also be included into alternative classification schemes. Suggestions range from the idea that NMD may be a whole family of mechanisms whose members each use a different set of NMD effectors and RNA features, to the surmise that there is no such thing as a dedicated NMD pathway (see below).

Many if not most NMD substrates do not have nonsense or premature termination codons and—at least in metazoans—no precisely defined NMD pathway could be identified. This discovery led to the suggestion to operationally define the “phenomenon commonly called NMD” as “translation-dependent mRNA degradation that requires UPF1 and in metazoans additionally SMG1 and SMG6 or SMG7” [226]. However, this characterization also applies to TRIM71-MD and maybe UPF1/SMG7/miRNA-MD. Therefore, these UMDs are either sub-forms of NMD or the selected criteria do not reliably and sufficiently distinguish NMD from other decay pathways.

Hug and colleagues [227] have proposed subdividing NMD into “strong” and “variable” NMD. Strong NMD acts on faulty transcripts such as transcripts with nonsense mutations or resulting from faulty splicing and other gene expression errors. Such transcripts would contain at least one PTC and a downstream normal termination codon. Strong NMD removes deleterious transcripts and functions as quality control. Non-functional strong NMD is responsible for disease phenotypes. By contrast, variable NMD acts on normal error-free transcripts and is elicited by a combination of weaker features such as uORFs, long 3′UTRs, exon junctions after a normal stop codon and certain sequence elements. It serves to fine-tune the expression of endogenous transcripts and to regulate the downstream pathways of proteins encoded by these transcripts. This is an appealing concept which helps to create a mind-map into which the various known features of NMD and probably other UMDs can be assimilated. A similar suggestion differentiates between “traditional” and “probabilistic” NMD substrates [105]. Traditional substrates comprise those with nonsense mutations, uORFs, splice errors, programmed frame shifting and pseudogenes. By contrast, probabilistic substrates are generated by leaky scanning, transcription start site errors and unprogrammed frameshifting (see below).

Other suggestions offer explanations for the surprisingly broad range of NMD efficiency or introduce a time dimension into NMD target recognition [78,141]. NMD rids cells of faulty RNAs thus preventing the expression of potentially harmful proteins. However, NMD also sophistically finetunes the expression of sometimes rare - yet important - transcripts, regulates whole pathways and contains an autoregulatory feedback loop [228]. Too aggressive quantity and quality control may do more harm than good. Therefore, overall NMD efficiency has been suggested to be a compromise of balancing both efficiency with specificity and quality control with regulation [8]. Against the background of cell type and physiological situations, NMD thus specifically maintains a regulatory flexibility weighing the abundance of targets versus nontargets, the abundance of NMD factors and the abundance of feedback inducing proteins and mRNA features [8,228,229].

An early model integrated the assumption of slow and inefficient termination at a PTC caused by an improper mRNP environment with UPF1′s ATP hydrolysis rate functioning as a molecular clock to determine the fate of an mRNA by a kinetic proofreading process [141]. This concept has recently been rekindled by the discovery that UPF1 is enriched on 3′UTRs after translation of the ORF and that target discrimination includes both UPF1 phosphorylation on substrate mRNAs and ATPase-dependent and PTBP1-promoted release of UPF1 from non-target mRNAs [77,78,125,129].

A very radical and thought-provoking proposition posits that there is no such thing as NMD [104]. The authors use similar arguments as presented here regarding the dispensability of individual protein factors and RNA features for NMD and regarding the variability and unpredictability of NMD substrates and NMD efficiency in cells, tissues and organisms. Moreover, they correctly argue that NMD impairment as a consequence of the depletion or deletion of certain factors does not necessarily prove that these factors have a role in NMD. It could as well be that NMD impairment is a side effect of such manipulations. For example, a whole set NMD effectors have been reported to moonlight in telomere maintenance (UPF1, UPF2, SMG1, and SMG5-7) and DNA damage response (UPF1, UPF2, SMG1, SMG6, SMG7) (reviewed in [230]). Furthermore, UPF1 is associated with transcriptionally active gene loci and nascent (pre-)mRNAs [133,231] and UPF1 depletion causes cellular stress and (in flies) nuclear mRNA retention [133,232]. Thus, depletion of UPF1 or other NMD effectors may disturb the expression of whole sets of genes due to the interruption of a number of pathways unrelated to NMD. Moreover, in some cases, although depletion of NMD factors affect the expression of hundreds of genes, the resulting cellular or organismal phenotypes are surprisingly mild and more severe phenotypes (or lethality) can sometimes be attributed to the overexpression of a single gene [104]. Consequently, Brogna and colleagues suggest that any mRNA with long regions unprotected by scanning ribosomes is degraded not by a specific surveillance mechanism, but by default [104], thus resuming a proposal made as early as 1999 [141]. Consistently, many NMD substrates display a lower ribosome density than non-substrates [233,234]. The function of UPF1—and possibly UPF2 and UPF3 at least in yeast [68]—is restricted in this model to a role in ribosome release and in resolving 3′UTR secondary structures by UPF1′s helicase function. Both functions would not be NMD-specific, but apply to all mRNAs with unprotected regions [104,235]. An appealing complementation of this model results from the discovery that ribosomal frameshifting occurs normally on many mRNAs with NTCs, thus constantly generating NMD substrates probably in each round of normal translation [233] and explaining why many mRNAs with no recognizable NMD features are directed to decay. In accordance with the “no-NMD-model”, such mRNAs have longer than average stretches of non-optimal codons with a lower density of ribosomes. Therefore, NMD can be “considered as a probabilistic mRNA quality control pathway that continually monitors errors affecting the maintenance of the normal reading frame [and] is active throughout an mRNA’s life cycle” [105].

## 24. A New Metaphor to Understand the UMD Family

Science needs metaphors to develop and communicate theories about natural phenomena. Vice versa, the metaphors chosen at any given time period shape our minds and our ways of thinking. Metaphors by no means replace biochemical, biophysical or kinetic calculations. They precede and surround these rigorous scientific instruments and help to form or understand hypotheses.

The longest-lasting and most powerful metaphor used in science is the scripture metaphor, starting with the perception of nature as a book that can be read, understood and interpreted. To this day, scripture metaphors are very popular in molecular biology as exemplified by terms such as message, transcription, translation, copying, editing and deciphering codes. However, interestingly and despite many efforts made [236], the genetic “code” could not be solved by cryptographers or linguists, but only experimentally. Sigmund Freud, being aware of their power, borrowed his metaphors from the mechanic industry of his time as exemplified by his concepts of the mental apparatus and others whose mechanical ancestry is only obvious in his mother tongue, such as *Druck* (pressure), *Trieb* (drive, instinct) and *Verdrängung* (displacement, repression). Industry metaphors are still popular in modern molecular biology: we talk about assembly lines, transport or transport chains and molecular machines. More recently, information and communication technology-related metaphors have mingled with the industry metaphors as exemplified by the use of terms like information storage and transfer, networks and crosstalk. Other (and partly overlapping) metaphors popular in cell and molecular biology are borrowed from the realms of social organization and administration (hierarchies, differentiation, specialization, development, control, stability, management, resources, access, etc.) or of games (players, teams, competition). All these metaphors have been very fruitful both regarding the formation and the communication of biologic concepts.

Theories—and the metaphors tied to them—sometimes last a very long time. With time, though, inconsistencies start to build up in every theory–metaphor relation. This process can take quite a while, until the theory and with it the metaphors that support and transport it become inadequate, signaling that it is time for a paradigm shift [237].

It seems that in the NMD field we are slowly approaching such a paradigm shift. However, over several decades of research a wealth of details about NMD has been accumulated, we have no satisfactory concept to accommodate these details. It seems that the more we know, the less we understand how all these jigsaw pieces can be arranged to form a complex, but coherent image. To understand NMD we are currently using a mixture of industry and information technology metaphors. We talk about the assembly line that shapes the EJC at splice junctions, about disassembly or recycling of post-termination ribosomes, about branched NMD pathways, dynamic networks of NMD effectors, UPF1 as molecular motor or molecular clock, etc. These very powerful and fruitful metaphors will be with us for a long time. However, it is time to complement them with new ones that we—as always—can borrow from current trends. One that we propose to test for manifestations of UPF1-mediated decay is the computing cloud as a metaphor for providing a flexible infrastructure with rapid elasticity and dynamic access according to specific user needs.

The unifying feature of all UMDs is UPF1 bound to RNA and interacting with either RNA structures, other RNA-bound proteins, or both. UPF1 appears to be bound to 3′UTRs possibly with a slight preference for G/GC rich sequences, which may indicate that translocating UPF1 is stalled by RNA or RNP structures thus enhancing its own local concentration [123] and that of other effectors by recruiting them and forming local foci of decay promoting effectors. Accordingly, the UMD world can be envisioned as a system where an initial “condensation nucleus” encompasses UPF1, some sort of RNA/RNP structure and maybe one or few pathway-specific UPF1 interactors. This condensation nuclei (red dots in Figure 3) trigger the recruitment of further effectors from a cloud-like pool whose components and molecular abundancies may differ in different cell types, physiological situations, or stages of development. A versatile sequence of events makes UMD ever more likely to happen [8]. This sequence includes UPF1 binding or recruitment to an RNA (region), activation of its ATPase and/or helicase functions, dissolution of three-dimensional RNA structures, stalling, phosphorylation and recruitment of a flexible set of UMD promoting factors. A cloudlike common pool of resources provides rapid and flexible access to the factors needed. The “on-demand” selection of UPF1 functions and of accessory factors from the “cloud” by the “user” (the specific UMD) determines how an RNA will be degraded. A strictly definable NMD pathway probably does not exist. NMD or rather the various NMD branches and UMDs, are only some of many dynamically appearing and fading manifestations of a polymorphic UPF1 cloud (Figure 3).

While this image does not explain every gap in our understanding of NMD and other UMDs, it may help to shape new ideas and develop new testable hypotheses. However, we should be aware that the human mind inevitably needs both, structured concepts and metaphors. Nature is different.

## Figures and Tables

**Figure 1 biomolecules-10-00999-f001:**
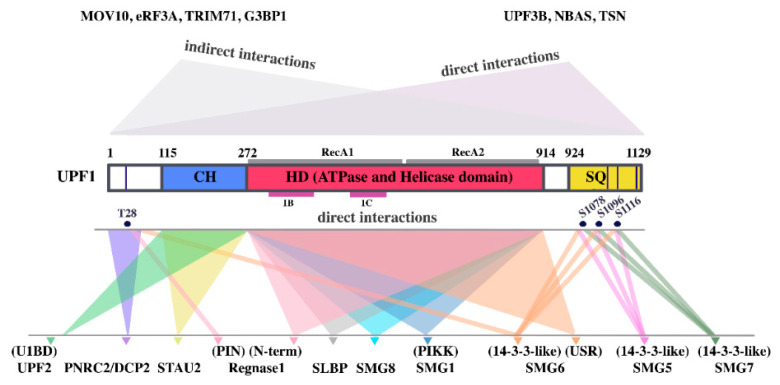
Functional up-frameshift factor 1 (UPF1) domains and their interactions with other proteins. СH: cysteine–histidine-rich domain, HD: helicase and ATPase domain with the RecA1, RecA2, 1B and 1C subdomains indicated by gray and magenta lines, respectively; SQ: serine–glutamine-rich domain. Validated phosphorylated amino acids with a function in protein–protein interactions, are shown in dark blue circles. Experimentally validated binding regions of UPF1-interacting UMD effector proteins are indicated below UPF1. UPF1-binding domains of UPF1-interacting proteins are indicated above the protein names. Proteins, that bind to UPF1 at as yet unknown sites are listed in the upper right corner. Proteins, that interact with UPF1 indirectly via RNA or other proteins are listed in the upper left corner.

**Figure 2 biomolecules-10-00999-f002:**
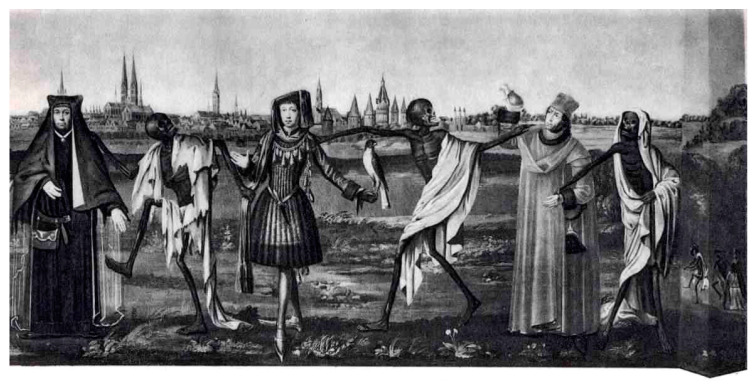
Detail of the *Danse macabre* (Totentanz) in the church St. Marien in Lübeck, Germany (Bernt Notke; ~1460; now destroyed). By analogy to this 15th century type of imagery, in the UMD family of decay pathways UPF1 represents Death (the skeleton in the picture) who leads the various types of UMD targets (representatives of any age and rank) to their final destiny.

**Figure 3 biomolecules-10-00999-f003:**
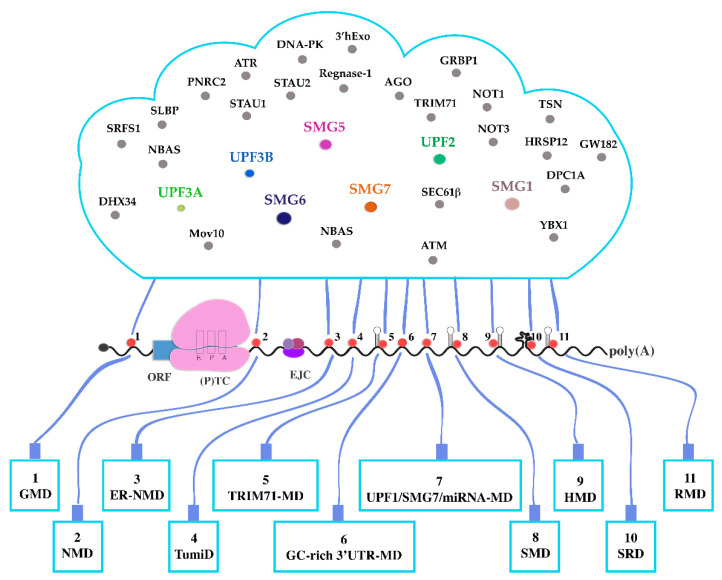
UPF1 “cloud”. The functioning of UMDs is compared to a network of users who all have access to a common “cloud”. UPF1 binds to or migrates to RNA stretches or structures. Depending on the UMD, binding is supported (or not) by other protein effectors, thus forming “condensation nuclei”. Pathway-specific condensation nuclei attract other effectors from a common pool. This pool is analogous to a computing cloud which contains shared and specific resources available on demand and according to the needs of the individual user (UMD). The UPF1-containing condensation nuclei are shown as red circles. UMDs are depicted as “end devices” connected to the “cloud” by “cables” through the condensation nuclei. Shared NMD-factors in the cloud are highlighted by colors. Other effectors that are specific for individual UMDs are shown in gray. Pathways 5, 8, 9 and 11 require dsRNA structures, shown as hairpins. Pathway 10 requires complex secondary 3′UTR structures shown as a multiloop knot. Most UMDs imply UPF1 interaction with 5′ or 3′UTRs. In the TumiD (pathway 4) UPF1 functions by dissociating miRNAs from target mRNAs. Most known UMDs (except 1 and 4) depend on translation termination at normal (TC) or premature (PTC) termination codons. The presence of an exon junction complex (EJC) considerably enhances NMD efficiency. Black curved line: RNA. Blue rectangle: ORF. Pink bipartite structure: terminating ribosome with the E, P and A sites; black circle: 5′ cap; poly(A): Poly(A)-tail.

**Table 1 biomolecules-10-00999-t001:** Features and requirements of UPF1-mediated RNA decay pathways.

UMD	RNA Target	NMD Effectors	Translation Termination	Selected Other Effectors ^1^	UPF1 Functions		UPF1 Phosphorylation	(p)-UPF1 Binding Factor
					ATPase	Helicase		
NMD	mRNAs, (lncRNAs; snoRNA hosts)	UPF1+ (in variable combinations): UPF2, UPF3A, UPF3B, SMG1c, SMG6, SMG5/7	yes	EJC, Mov10, DHX34, SRSF1, NBAS, long 3′UTR decapping and deadenylation factors, 3–5′ and 5′–3′ exonucleases, PNRC2	yes	yes	yes	RNA?
SMD	mRNA	UPF1, UPF2, SMG1	yes	STAU1, STAU2, dsRNA structures, PNRC2	yes	yes	yes	UPF2 (STAU)
HMD	Histone mRNA	UPF1, SMG1	yes	RNA stem loop SLBP, ATR, DNA–PK, 3‘hExo, PNRC2	yes	yes	yes	3′UTR of histone mRNAs
RMD	Proinflammatory cytokine transcripts	UPF1, SMG1	yes	Stem loop, with Py–Pu-Py-loop, regnase-1	yes	yes	yes	regnase-1
SRD	mRNAs, circular RNAs	UPF1	yes	GRBP1	yes	yes	n.d.	Structured RNA
TRIM71-MD	CDKN1A/p21 mRNA	UPF1, SMG1, SMG7	n.d.	TRIM71, TRIM71-binding stem loop in 3′UTR	n.d.	n.d.	implied ^2^	TRIM71?
GC-rich 3′UTR-MD	mRNAs with GC-rich 3′UTRs	UPF1, UPF2, SMG1	n.d.	n.d.	implied	implied	implied	GC-rich mRNA
UPF1/SMG7/miRNA-MD	mRNAs with CUG sequences embedded in miRNA seed sequences in 3′UTR	UPF1, SMG7,	n.d.	miRNA loaded AGO, NOT1, NOT3	n.d.	no	n.d.	CUG embedded in 7mer miRNA seed sequence
TumiD	miRNA	UPF1	no	Tudor staphylococcal/ micrococcal-like nuclease (TSN), AGO2, GW182	n.d.	yes	No (by implication ^2^)	TSN
GMD	mRNA with GR-binding site often in 5′UTR	UPF1	no	glucocorticoid, glucocorticoid receptor (GR), ATM, PNRC2, DCP1A, YBX1, HRSP12	yes	yes	yes	GR-PNRC2 complex
ER NMD	ER-localized mRNAs	UPF1 (others?)	n.d.	NBAS, SEC61β	n.d.	n.d.	No (by implication)	NBAS

^1^ This list represents a selection of effectors as reported in the literature discussed in the text without claim to be exhaustive. ^2^ “implied” or “by implication“ indicates that the respective UPF1 function or feature was not tested directly but concluded upon by using UPF1 variants with mutations that interfere with these functions. N.d.: not determined.

**Table 2 biomolecules-10-00999-t002:** Exemplary mRNA targets of more than one UMD.

mRNA	NMD-Inducing Feature	2nd UMD	UMD-Inducing Feature
*PTGS2*	uORF	RMD	long structured 3′UTR
*ATF3*	uORF, PTC introduced by alternative splicing	RMD	long structured 3′UTR
*GADD45B*	3′UTR intron		long structured 3′UTR
*COL22A1*	PTC introduced by alternative splicing	RMD	long structured 3′UTR
*PEA15*	uORF	UPF1/SMG7/ miRNA-MD	miRNA seed sequence complementarity in 3′UTR
*NFKBIB*	PTC introduced by alternative splicing	RMD	long structured 3′UTR

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
