# Peer review of "UPF1-Mediated RNA Decay—Danse Macabre in a Cloud"

_biomolecules, 2020, doi:10.3390/biom10070999_

Round 1
Reviewer 1 Report
This review is well researched. It is clear that the authors have a great deal of knowledge regarding these topics and this would be of value to the field.
My major criticisms are that the organization and writing lack clarity and should be improved before final publication. The English has no problems, but, it seems as the authors have prioritized eloquence at the expense of clarity. There are several instances of sentences that are not technically run-ons, but have so many clauses, that the point of the sentence is lost. The first sentence is a prime example. The paragraph starting at line 360 with the phrase: “To this day, NMD-related publications state that…” is exceeding difficult to comprehend. There is much to be appreciative in this manuscript and I worry that much will be lost to a casual reader.
The authors provide excellent details on individual mechanisms, but need to provide better background so that this will be more useful to people not immersed in the NMD or RNA decay field. For example, certain proteins are described, but never introduced. That is, they aren’t described for a novice. For example, the SMG proteins are poorly described. The 1st time they are mentioned in the text is with the sentence: “Phosphorylation of the N-terminal T28 site and the C-terminal S1096 and S1116 are crucial for SMG6 binding and for SMG5 and/or SMG7 binding, respectively.” Prior to this, the SMG proteins should be better introduced. Throwing these names at some people means little. It a couple sentences later that SMG6 is identified as “endonucleolytically cleaving” RNA. It would be better to start with saying that SMG6 is a PIN-domain containing endonuclease that was identified… Then describe how it is recruited. The same is true for 5/7.
Similarly, the EJC is discussed much, but is never described. The authors should give some brief input into the components of the EJC and how they are deposited.
Also, there are times when this article comes off more as a “perspective” or “commentary” rather than a review. Some degree of editorializing and critiquing of data can improve a review, but there are times here, where the authors have crept right up to the line. I believe that the final two sections are appropriate, but earlier, editorializing is interspersed with description making it difficult to determine if what is stated is fact or opinion. The role of Upf2 in NMD is a prime example of this problem.
The authors rely heavily on citations of other reviews. They should try to highlight more primary literature where appropriate.
Some statements are too declarative when there is some controversy surrounding a topic. Rather, the authors should highlight some of the controversies. For example, stating that “UPF1 does not interact with release factors” is a very declarative statement in contrast to many years of data showing otherwise. It is true that recent data challenges this, but it does not nullify previous work.
While an admittedly difficult subject, the authors should similarly approach the “pioneer round” of translation. Given the amount of work published on this topic, it cannot be dismissed in a 5 sentence paragraph.
Why is “Regnase-1 dependent UPF1-mediated mRNA Decay” a subheading under histone mRNA decay?
ATR has been shown to be required to histone mRNA decay upon inhibition of DNA synthesis (Kaygun & Marzluff). In table 1, the authors state state: (ATM?). Did the authors mean ATR? If not, ATR should be added. Also, they should provide a citation for the role of ATM in this process. If it was meant to be ATR, why is there a question mark and why is it in parenthesis?
The authors repeatedly refer to “GW418”. I presume this is meant to be GW182.
The sentence starting on line 632 regarding G3BP is very misleading. G3BP is not an accepted nuclease despite early papers (Tourriere 2001, 2003). Rather, it more likely functions as a “condesase”. See recent papers in Cell from Brangwynne, J. Paul Taylor and Franzmann Labs. Further, Fischer et al do not directly implicate G3BP as being an endonuclease, but only state a that an endonuclease is likely involved while refereeing the previous Tourriere papers. Additionally, the role of S149 phosphorylation has recently been disputed as it likely arose from a sequencing error, See Tourriere & Tazi, 2019 and Panas et al. 2019 JCB. These sentences should be reworded as to not give such a strong implication that G3BP is a nuclease. It certainly has a role in this pathway along with Upf1, but is unlikely to harbor nuclease activity.
Line 155: (Isken 2008) is in the text. I suppose this was a placeholder, but should be updated in the reference section.
Line 312: The authors refer to RNA “circularization”. Instead, the authors should say “closed-loop” formation, particualry as circular RNAs become more in vogue.
Some references are duplicated. For example, ref # 24 and 30 are the same paper.
Reviewer 2 Report
Lavysh & Neu-Yilik provide a comprehensive and intriguing review on the topic of RNA turnover mechanisms revolving around the RNA-binding protein UPF1. The authors first give an extensive description of NMD, the currently best understood RNA turnover pathway in the field. They provide an excellent discussion of NMD mechanism and models. They then introduce and describe in detail a whole host of other UPF1-dependent “mechanisms” that were defined by different groups, each of which was christened with a different acronym by the original discoverer; e.g., SMD, GMD, etc. This information, while written in an encyclopedic manner and thus “hard-going” at times, will be extremely valuable for the field, particularly since it is all in one place. Table 1 will be very useful for the field. The authors then go on to reflect on what all this wealth of information about different UPF1-dependent pathway really means. They provide a balanced view of various explanations, including that all these pathways are really one pathway. Indeed, they provide a convincing case that many (or even all) of the pathways are really variations on a single pathway. They even go over the intriguing hypothesis from Brogna that NMD is not really a pathway but instead an artifact of NMD factor knockout and knockdown experiments. A key point they make is that no single NMD factor except UPF1 or single RNA features is sufficient or indispensable for all NMD. The authors finish by bringing up the power of metaphors in science and then propose their own metaphor that collectively explains the diverse array of UPF1-dependent RNA turnover pathways. This is not a standard review article, which I consider to be its strength.
While I am enthusiastic about this review being published, there are many issues that the authors should attend to:
- I really like the phrase “danse macabre in a cloud,” but I don’t think that it belongs in the title. Before readers can appreciate the meaning of this poetic phrase, they will need to be told what “danse macabre” means and they will need to be properly introduced to the “cloud” metaphor meant by the authors. This line (or some form of it) should instead be the last line in the Abstract and/or in the body of the review. It is suggested to instead have a more “functional” title with key phrases that readers interested in the topic of RNA will catch. This will increase readership and impact.
- Danse macabre. This dance of death metaphor is certainly fitting for regulated RNA decay and thus it is critical that it be defined (currently, do definition is provided). Also, the authors should explain how this phrase nicely captures the mechanics and other aspects of RNA decay. It is also strongly suggested to include a fig or fig panel depicting danse macabre.
- Line 28. Since NMD also degrades non-mRNAs, the authors should define it as “nonsense-mediated RNA decay.”
- Line 30. Some 5’UTR regions can also trigger NMD.
- Awkward writing. One of many examples of this is on line 50, where “not seldom” should be replaced with a word such as “often.” Unfortunately, the manuscript is filled with such awkward phrases or whole sentences (I only cover a couple more below). This will probably not be a problem for experts in the field, but it is suggested to make improvements in the writing for non-RNA aficionados. The best plan would be to get editing help from either an English major or a scientist whose native language is English (I apologize in advance if either of the authors fits either of these categories).
- Line 69. This assumes that expression patterns are always due to purifying selective forces, which is certainly not always true. Most genes are probably expressed in some places and at times where they are not needed. Unless there is selection pressure to remove something (e.g., because it is toxic), often it will persist over evolutionary time. This is something to keep in mind throughout the review.
- Line 112. No citation provided.
- Section titles on lines 146 & 189. “of,” not “in.”
- Sections 4-10, each of which cover a different topic about NMD, seem disparate. Each section brings up an important issue in the field, so I’m all for the content, but there is little or no linkage between the sections. The authors should also think about the possibility of re-organizing the sections to make them more seamless.
- Line 286. No citation.
- Section 9. The authors should discuss the model supported by some literature that UPF1 is recruited to all mRNAs but has a slow off-rate specifically on NMD targets.
- Section 12.1. “Regulatory RNase I” is not defined at the beginning and is defined different ways subsequently.
- Line 423. It is not explained until later what “suppress RNAs” means. This could refer to rapid RNA decay, translational silencing, as well as other mechanisms. This kind of problem is elsewhere in the review, including on line 552 - “repressor” of what stage of gene expression?
- Line 547. Rephrase so “intriguingly” is explained.
- Section 15. Explain the pathway before getting into the details, including how predominant it is. This is a general problem throughout the review. Think carefully about what flow of information works best. In general, follow the “funnel principal,” where one starts general and then gets more specific.
- Line 592. “Posit,” rather than “state.”
- Line 634. Is “UPF1” meant here?
- Line 722. “both” is confusing, as readers may not have noticed that two species were mentioned in the topic sentence.
- Line 795. The authors bring up the key point that “NMD efficiency has been suggested to be a compromise of balancing both efficiency with specificity and quality control with regulation.“ It is suggested to expand further on these ideas, as these have huge implications for both how NMD evolved and how it functions in the present day.
- Line 872. As currently written, the authors briefly bring up their “computing cloud” metaphor and then swiftly change subject They should instead explicitly explain how the internet cloud works and then point out how each of these features relate to UPF1-mediated RNA turnover. It is strongly suggested to provide an explanatory figure. 2, while ok, doesn’t depict the specific analogous features between RNA turnover control and the cloud.
Reviewer 3 Report
“UPF1 mediated RNA decay –danse macabre in a clouhd” from Lavish et al is a very interesting and useful review discussing the multiplicity of the RNA decay pathways relying on the DNA/RNA helicase UPF1. The prototypical process is the nonsense mediated mRNA decay (NMD). It is first described according to the 2 historical and prevailing models. More interestingly, the authors then report the gaps and inconsistencies of these models by raising 6 critical points. Finally they questioned the indispensable nature of the so called NMD core factors and cis-acting features controlling NMD. They conclude that except UPF1, other factors are neither indispensable nor sufficient to trigger NMD. After that analysis, the authors listed and described all the UPF1 mediated mRNA decay pathways and try to define the conceptual difference between these pathways in order to propose a strictly definable NMD if possible.
Although the organisation of the review leads to redundancies that the authors should decrease at the maximum (by condensing the paragraph1-3 for instance), it is very pleasant (because it’s rare) to read a review that succeed to question the dogma of NMD mechanism and not just states the findings from each article. At the light of what has been written in the previous paragraph, the concluding paragraphs (22 and 23) seem repetitive and should be condensed as well.
Specific comments:
Paragraph 14: I wonder if the GC rich 3’UTR really can be considered a UMD process? To my opinion the whole paragraph is just a development of the paragraph 10.4 about RNA features modulating NMD.
Paragraph 20: the crosstalk between UMD and non UPF1 related RNA decay pathways is interesting but out of the scope of this study, to the contrary of the paragraph 21.
Paragraph 23: I am doubtful on the metaphoric paragraph of this review.
Minor comments:
P2 lane.73 bracket missing
P3 lane101: the comma should be removed between DHX34 and MOV10.
P3 lane.113: I am not convinced that this paragraph is necessary for the following demonstration.
P4 and P5: why inverting the decay phase of NMD and the translation termination and not follow the chronological order?
P5 lane.215 “question” should be “questions”
Problem with the page labelling after page 10 it goes back to page 2.
Reviewer 4 Report
Lavysh and Neu-Yilik provided a nice review summarizing a variety of mRNA decay pathways, all of which are dependent on UPF1. Overall, there are a few points for the authors to consider and revise for clarity.
- line 181: Although Nicholson et al., (RNA. 2018) showed that PNRC2 weakly elicits degradation of NMD substrates in comparison with SMG5/7, this group clearly showed that a tethered PNRC2 to 3’UTR of reporter mRNA strongly triggers reporter mRNA degradation in a stop codon-dependent manner. On the other hand, Cho et al., (Nucleic Acids Res. 2013) showed that SMG5-PNRC2 is functionally predominant compared with SMG5-SMG7 at the transcriptome level. Therefore, it is clear that PNRC2 is involved in NMD. A debating issue is whether PNRC2 is a general factor for NMD.
- line 231: “the cap-binding factors play no role in the discrimination……” This sentence is misleading. It was shown that both CBC and eIF4E are comparably engaged in NMD in yeast (ref. 97). In contrast, in mammalian cells, CBC promotes NMD via its interaction with UPF1 (Hwang et al., Mol. Cell, 2010). At least in mammalian cells, CBC has some active roles in NMD.
- Table 1: PNRC2 is involved in many UMDs, such as NMD, SMD, HMD, and GMD. In the present table, it looks like that PNRC2 is a specific factor for GMD, which is misleading. Therefore, this reviewer recommends that the authors consider to add PNRC2 to other effectors involved in NMD (Cho et al., Mol Cell, 2009), SMD (Cho et al., Mol Cell, 2012), HMD (Choe et al., Nucleic Acids Res. 2014), and GMD (Cho et al., Proc. Natl. Acad. Sci. 2015).
- line 518: is there any reason why the number is 12.1 rather than 13?
- Throughout the manuscript, it would be better if the nomenclatures are consistent. For instance, Reg1-UMD vs RMD and GR-UMD vs GMD. Readers may easily understand that all decay pathways described in this manuscript are dependent on UPF1. Therefore, this reviewer recommends that the authors simply follow previously reported nomenclatures (NMD, SMD, GMD, and RMD).
- line 669: “requires UPF1 recruitment to the 5’UTR instead of the 3’UTR”. GMD triggers rapid degradation of its substrates independently of a position of GR-binding site, 5’UTR or 3’UTR (Cho et al., Proc. Natl. Acad. Sci. 2015).
- line 680: Although HRSP12 was annotated to endoribonuclease, it is recently shown that a purified HRSP12 lacks any ribonuclease activity in vitro system (Park et al., Mol. Cell, 2019). It would be better if the above information is added.
- line 690: looks like a typo.
Round 2
Reviewer 1 Report
The revisions have resulted in an even better paper. This paper will be of great value to the field.